# Cobalt Phosphotungstate-Based Composites as Bifunctional Electrocatalysts for Oxygen Reactions

Ndrina Limani [1,2], Inês S. Marques [1], Bruno Jarrais [1,*], António J. S. Fernandes [3], Cristina Freire [1] and Diana M. Fernandes [1,*]

1 REQUIMTE-LAQV/Departamento de Química e Bioquímica, Faculdade de Ciências, Universidade do Porto, 4169-007 Porto, Portugal; ndrinalimani@gmail.com (N.L.); up201608306@edu.fc.up.pt (I.S.M.); acfreire@fc.up.pt (C.F.)
2 Universite Paris Saclay, CEA, CEA Saclay, CNRS, NIMBE, LICSEN, F-91191 Gif Sur Yvette, France
3 Departamento de Física, Instituto de Nanoestruturas, Campus Universitário de Santiago, Nanomodelação e Nanofabricação (I3N), Universidade de Aveiro, 3810-193 Aveiro, Portugal; toze2@ua.pt
* Correspondence: bruno.jarrais@fc.up.pt (B.J.); diana.fernandes@fc.up.pt (D.M.F.)

**Abstract:** The oxygen reduction reaction (ORR) and oxygen evolution reaction (OER) are key reactions in energy-converting systems, such as fuel cells (FCs) and water-splitting (WS) devices. However, the current use of expensive Pt-based electrocatalysts for ORR and $IrO_2$ and $RuO_2$ for OER is still a major drawback for the economic viability of these clean energy technologies. Thus, there is an incessant search for low-cost and efficient electrocatalysts (ECs). Hence, herein, we report the preparation, characterization (Raman, XPS, and SEM), and application of four composites based on doped-carbon materials (CM) and cobalt phosphotungstate (MWCNT_N8_Co4, GF_N8_Co4, GF_ND8_Co4, and GF_NS8_Co4) as ORR and OER electrocatalysts in alkaline medium (pH = 13). Structural characterization confirmed the successful carbon materials doping with N and/or N, S, and the incorporation of the cobalt phosphotungstate. Overall, all composites showed good ORR performance with onset potentials ranging from 0.83 to 0.85 V vs. RHE, excellent tolerance to methanol crossover with current retentions between 88 and 90%, and good stability after 20,000 s at $E = 0.55$ V vs. RHE (73% to 82% of initial current). In addition, the number of electrons transferred per $O_2$ molecule was close to four, suggesting selectivity to the direct process. Moreover, these composites also presented excellent OER performance with GF_N8_Co4 showing an overpotential of 0.34 V vs. RHE (for $j = 10$ mA $cm^{-2}$) and $j_{max}$ close to 70 mA $cm^{-2}$. More importantly, this electrocatalyst outperformed state-of-the-art $IrO_2$ electrocatalyst. Thus, this work represents a step forward toward bifunctional electrocatalysts using less expensive materials.

**Keywords:** oxygen reduction; water oxidation; carbon materials; N; S-doping; polyoxometalates

## 1. Introduction

Global population numbers doubled since the 1970s to approximately 8 billion and will almost certainly continue to grow to a staggering 10 billion individuals still within this century [1]. The energy demand of this growing population is also increasing at an alarming rate, and the present heavy dependence on fossil fuels is causing a staking anthropogenic climate change [2,3]. To reverse this tendency, mankind must move away from fossil fuels combustion and toward renewable, clean energy sources. Among the most popular clean energy devices are fuel cells (FCs), metal–air batteries, and water-splitting devices. However, the oxygen reduction sluggish kinetics at the FCs and metal–air batteries and the high overpotentials required for oxygen evolution at the water-splitting devices limit their large-scale application [4]. Thus, efficient ORR and OER electrocatalysis is crucial to the implementation of these devices regarding the sustainable production/storage of energy in the future. This has stimulated the search and development of new electrocatalysts over the last decade.

For ORR, platinum-based nanomaterials have shown leading results so far, while for OER, the state-of-the-art electrocatalysts are $IrO_2$ and $RuO_2$ [5], all of them being scarce and having a high cost, which makes them not suitable for large-scale applications.

Multiple carbon nanomaterials such as graphene, carbon nanotubes, and metal–organic frameworks (MOF) derived nanocarbons have been explored for their implementation as electrocatalysts for ORR and OER considering their low cost, high surface area, great conductivity, and high stability [4,6–10]. Nitrogen-doped carbon nanomaterials have shown to enhance even more the electrocatalytic activity toward ORR and OER, possibly by the electron density adjustment from the nitrogen dopant [11,12]. Several types of nitrogen atoms that can be introduced in the carbon structure include pyridinic, pyrrolic, graphitic (quaternary), and oxidized nitrogen (Figure 1a) [13]. The co-doping of nitrogen and sulfur on carbon nanomaterials has also shown a positive synergetic effect toward these reactions [6,14].

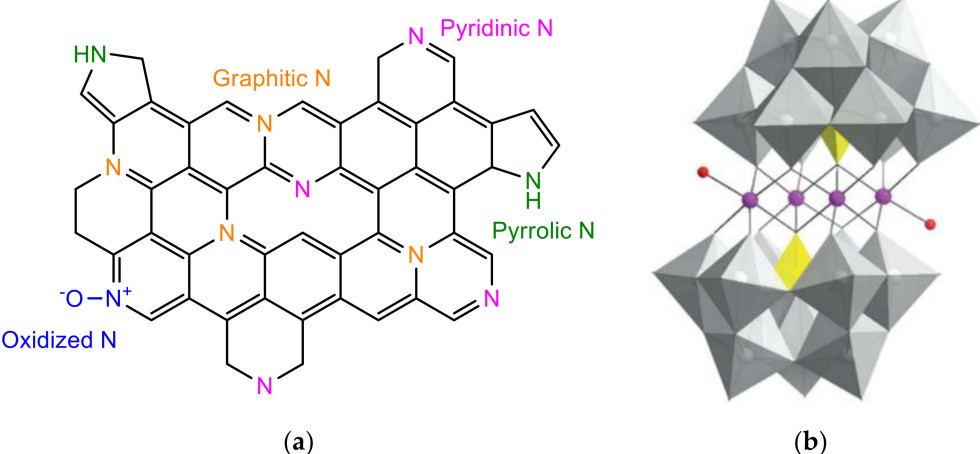

(**a**)　　　　　　　　　(**b**)

**Figure 1.** Schematic of N-types in graphene structure (**a**) and sandwich-type POM (**b**).

Moreover, the design and preparation of composite electrocatalysts by integrating the excellent properties of carbon nanomaterials with the high stability and good redox properties of polyoxometalates (POMs) has been reported as a possible successful way for enhancing the efficiency toward different energy-related reactions (ORR, OER, and HER) [4,15–17]. POMs are inorganic molecular clusters based on metal–oxygen bonds, containing metal centers or addenda atoms (M) in their high oxidation states, different types of oxygen atoms, and heteroatoms [18,19]. They have manifested high affinity toward carbon supports through electrostatic interactions considering their anionic nature [20,21]. Due to their capability of transferring several electrons from the addenda atoms [22], POMs and in particular Co-based ones have been used as promising candidates for ORR and OER [23–28]. Several studies have been published by Galán-Mascarós and Poblet regarding the use cobalt polyoxometalates for water oxidation [25,29,30]. Stracke and Finke have also investigated the use of $Co_4(H_2O)_2(PW_9O_{34})_2]^{10-}$ (Co4, Figure 1b) as water oxidation electrocatalysts [31,32]. Hill et al. [27] have also shown that Co4 is a molecular water oxidation catalyst despite the studies by Stracke and Finke [31] that showed a decomposition of Co4 to CoOx under electrochemical bias. However, their application alone as electrocatalysts has been hampered by their low surface area of only several $m^2$ $g^{-1}$, high solubility in polar solvents, and negligible conductivity [33,34].

With this in mind, in the present work, we have explored the preparation of different N- and N, S-doped carbon materials to serve as bridging atoms to facilitate electron transport and the subsequent preparation of composites based on the previously doped carbon materials and a cobalt-based POM $[Co_4(H_2O)_2(PW_9O_{34})_2]^{10-}$ to act as electrocatalysts for both the electrochemical oxidation and reduction in oxygen. The materials were prepared

by a simple and scalable strategy without the need of linker molecules, which is per se an advantage.

Additionally, the nanocomposites benefited from both the heteroatom doping and the presence of POM exhibiting good ORR activity and superior OER electrocatalytic activities.

## 2. Experimental Section

### 2.1. Materials and Characterization Methods

The reagents and solvents used during the experimental execution of this work were used as received. Multi-walled carbon nanotubes (sample denoted as MWCNT) were commercially obtained from Nanocyl S.A., Sambreville, Belgium, Ref. 3100 MWCNT (>95% carbon purity; 9.5 nm average diameter). Commercial graphene (sample denoted as GF) was from Graphene Technologies (Lot #GTX-7/6-10.4.13, Novato, USA). Dicyandiamide (99%), sodium tungstate dehydrate (>99%) and sodium phosphate monobasic dehydrate (>99%) were from Sigma-Aldrich, Algés, Portugal. Dithiooxamide (≥98%) and potassium chloride (99.5%) were from Merck, Darmstad, Germany. Potassium carbonate (98%) was from VWR Chemicals, Amadora, Portugal. Melamine was from Alfa Aesar, Kandel, Germany (99%). The $N_2$ used in the modification of the carbon materials was from Praxair, Porto, Portugal (>99.998%).

For the electrocatalytic tests, we used the following solvents and chemicals: isopropanol (99.5%, Aldrich, Algés, Portugal), methanol (anhydrous, VWR, Amadora, Portugal), Nafion (5 wt % in lower aliphatic alcohols and water, Aldrich, Algés, Portugal), hydrogen peroxide solution (30 wt % in water, ACS reagent, Sigma Aldrich, Algés, Portugal), potassium hydroxide (KOH, 99.99%, Sigma-Aldrich, Algés, Portugal), and platinum nominally 20% on carbon black (Pt/C 20 wt %, HiSPEC® 3000, Alfa Aesar, Haverhill, MA, USA). Ultrapure water (18.2 MΩ cm, 25 °C, Interlab, Lisboa, Portugal) was used to prepare the electrolyte for ORR and OER studies. The pristine and modified materials were characterized by Raman spectroscopy (GF materials only), Fourier transform infrared (FTIR), X-ray photoelectronic spectroscopy (XPS), and scanning electronic microscopy (SEM). The detailed information about apparatus and methods is found in the Supplementary Materials file.

### 2.2. Materials Preparation

Synthesis of doped carbon nanomaterials: The incorporation of heteroatoms (N or N and S) onto the pristine carbon materials MWCNT and GF was realized through mechanical treatments in a ball milling Retsch MM200 equipment, with the appropriate heteroatom precursors (melamine for MWCNT_N8 and GF_N8, dicyandiamide for GF_ND8, and dithiooxamide for GF_NS8), followed by adequate thermal treatments under N2 flow. In a typical experiment, 0.60 g of carbon material was mixed with 0.26 g of doping element(s) using the appropriate precursor, and the mixture was ball-milled during 5 h at a constant frequency of 15 vibrations $s^{-1}$. Afterwards, the resulting materials were subjected to a thermal treatment under $N_2$ flow (100 $cm^3$ $min^{-1}$), at a rate of 10 °C $min^{-1}$ until reaching 800 °C, kept at that temperature during 1 h, cooled to room temperature under nitrogen atmosphere, and stored in a desiccator.

Synthesis of POM: $K_{10}[Co_4(H_2O)_2(PW_9O_{34})_2]$ (Co4) was synthesized as described previously [35]. Briefly, a mixture was prepared by dissolving $Co(NO_3)_2 \cdot 6H_2O$ (3.2 g, 11 mmol) in water (250 mL); afterwards, PW9 (15.0 g, 5.4 mmol) was added gradually through stirring for 30 min. Subsequently, the mixture was heated slowly until 50–60 °C and was kept at this temperature for 30 min while stirring. Then, filtration was performed and KCl (37.5 g) was added to the hot solution, which was kept stirring for 15 min. As a result, the formed violet-colored solid was filtrated and dried under vacuum.

Synthesis of composites: The composite materials were prepared as follows: 50 mg of doped-carbon material was dispersed in 50 mL of ultrapure water and stirred for 1 h, while separately, 25 mg of Co4 was dissolved in 12.5 mL of ultrapure water. Then, these two parts were mixed and stirred for 4 h followed by filtration under ultra-high vacuum. The black

final composite was finally dried overnight at 50 °C. For a better understanding of the abbreviation codes of all materials, please see Table S1 in the Supplementary Materials file.

### 2.3. ORR and OER Electrochemical Performances

Cyclic voltammetry (CV) and linear sweep voltammetry (LSV) tests were performed using a potentiostat/galvanostat Autolab PGSTAT 302N (EcoChimie B.V., Utrecht, The Netherlands), controlled by the NOVA v2.1 software. The cell used for electrochemical measurements consisted of 3 electrodes: a modified glassy carbon rotating disk electrode, RDE (Metrohm, Utrecht, The Netherlands, 3 mm of diameter) as working electrode, an Ag/AgCl (Metrohm, Utrecht, The Netherlands, 3 mol dm$^{-3}$ KCl) as the reference and a carbon rod (Metrohm, Utrecht, The Netherlands, 2 mm of diameter) for ORR or a platinum wire (d = 0.6 mm, 0.5 m, 99.99+%, Goodfellow) for OER as the counter electrode.

Before being modified, the electrode underwent a cleaning process (see the Supplementary Materials file for more details). To modify the RDE, a 5 μL drop of the selected material dispersion was deposited onto its surface and allowed to dry. The materials dispersion was prepared by mixing 1 mg of electrocatalyst with a solvent mixture of 125/125/20 μL of 2-propanol/ultrapure water/Nafion and dispersing using an ultrasonic bath for 15 min.

The ORR electrochemical experiments were performed in nitrogen and oxygen saturated KOH (0.1 mol dm$^{-3}$), while for OER studies, the electrolyte was only purged with nitrogen. The electrolyte was bubbled with the desired gas for at least 30 min. For simplicity, all the experimental conditions used, and the parameters evaluated for both reactions are described in detail in the Supplementary Materials file. The electrochemically active surface areas (ECSAs) for all materials were also estimated (see the procedure and theoretical comments in the Supplementary Materials file).

## 3. Results and Discussion

### 3.1. Materials Characterization

#### 3.1.1. Raman Spectroscopy

For the GF-based materials, Raman spectroscopy was used to study the structural changes introduced by the incorporation of the heteroatoms and POMs. The Raman spectra of these materials is presented in Figure 2 and is characterized by strong bands at ≈1350 and ≈1580 cm$^{-1}$, which are known as D and G bands, respectively, and a third band at ≈2700 cm$^{-1}$, which is assigned as the 2D mode, an overtone of the D peak. At ≈1620 cm$^{-1}$, it is also possible to observe a weak shoulder corresponding to the D' mode. A defect-free sp$^2$ carbon system would present the G band exclusively, assigned to the first-order scattering of the E$_{2g}$ mode, as the D band is due to a phonon mode due to the existence of six-fold aromatic rings close to local lattice distortions, or defects, of the graphitic network [36]. Such distortions are related to the presence of heteroatoms, edges of graphitic planes, atomic vacancies, or oxygenated groups. As such, the intensity ratio of the D and G bands has been commonly used to estimate the disorder degree of graphitic materials [37–40].

The intensity ratio of the D and G bands (ID/IG) can provide the degree of disorder and the average size of the sp$^2$ domains. Figure 3a presents the ID/IG ratio for the pristine and modified GF materials, and as can be seen, all doped materials present an increase in the ID/IG ratio, indicating that the doping procedures increased the amount of disorder in the doped materials and that the POM incorporation had a negligible effect on the graphitic structures.

The Raman spectra of the doped/modified graphene flake materials also present a very distinct feature in the form of a significant red shift of the D, G, and 2D bands. Figure 3b depicts the red shift of the D, G, and 2D bands of the doped/modified materials in comparison to pristine GF. This occurrence can be explained by the existence of tensile strain within the graphene flake materials. There have been several reports of red shifts in the Raman spectra of graphene materials as a result of tensile strain, with some authors even using it as a mapping tool for the accurate determination of such strain [41–43]. Therefore,

it is possible to conclude that the doping procedures induced non-reversible strain in the prepared doped graphene flake materials.

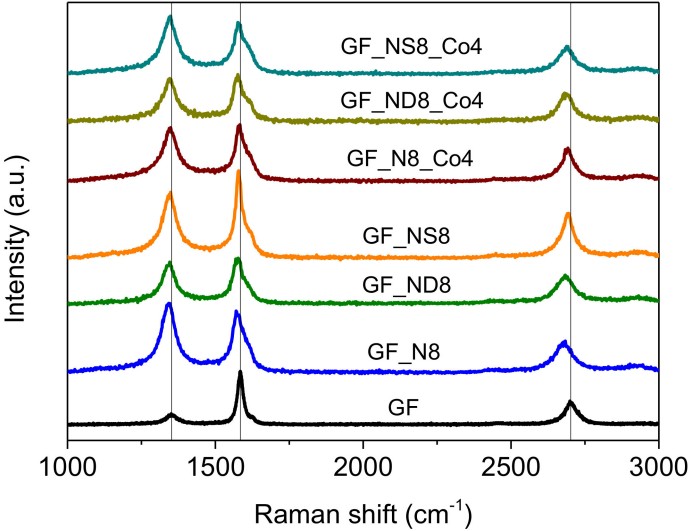

**Figure 2.** Raman spectra of the GF-based materials.

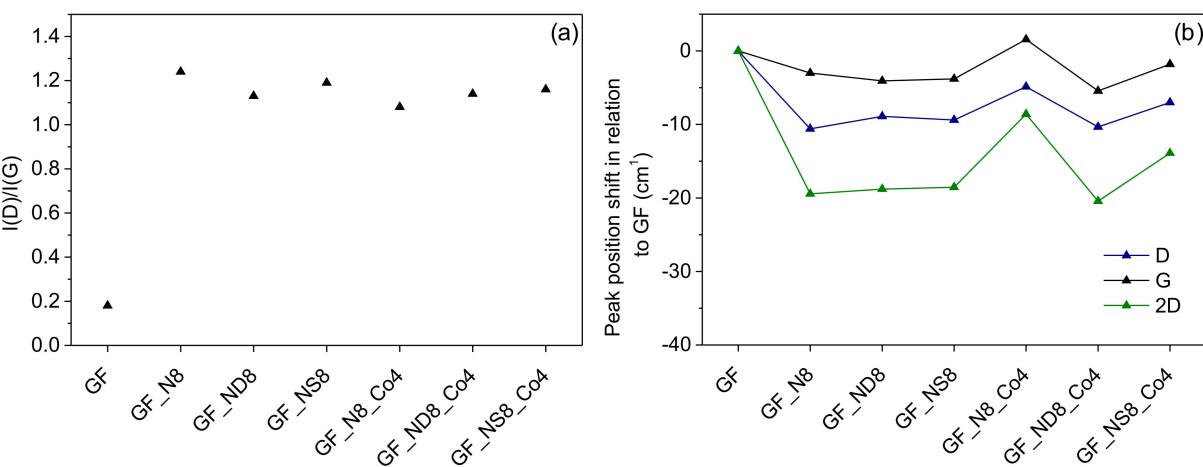

**Figure 3.** (**a**) Calculated ID/IG ratio for the pristine and doped/modified GF materials; (**b**) Raman red shift of the D, G, and 2D bands of the doped/modified materials in comparison to pristine GF.

### 3.1.2. Fourier Transform Infrared Spectroscopy

Since polyoxometalates have characteristic vibration bands in the frequency region between 1100 and 700 $cm^{-1}$, the FTIR spectra of $Co_4$ and the four composites prepared were characterized by FTIR, and the spectra can be observed in Figure S1. The Co4 FTIR spectrum shows the typical $P-O$ stretching vibration at 1039 $cm^{-1}$, the $W-Od$ vibration at 943 $cm^{-1}$, and the $W-Ob-W$ and $W-Oc-W$ stretching modes at 881 and 814 $cm^{-1}$, respectively [35]. The IR spectra of all carbon materials are similar and are shown in Figure S2. In the spectrum of MWCNT_N8, vibrational bands are observed at 3410 (OH groups stretching vibrations), 1568 (C=N or C=C stretching vibrations), 1384 (C−N stretching vibrations), and 1170 $cm^{-1}$ (C−O stretching vibration) [35,44]. In the spectra of doped graphenes, five bands are observed at 3431, 1638, 1577, 1388, and 1162 $cm^{-1}$ assigned to the stretching vibrations of OH groups, C=N, C=C, C−N, and C−O, respectively [44]. In the spectra of all composites prepared, we can clearly observe the vibration bands corresponding to the polyoxometalate, which confirms its immobilization.

### 3.1.3. X-ray Photoelectron Spectroscopy

All materials were analyzed by XPS in order to study their composition. The surface atomic percentages of each element for all materials are presented in Table 1.

**Table 1.** XPS relative surface atomic percentages for all materials [a].

| Sample | Atomic % | | | | | | |
|---|---|---|---|---|---|---|---|
| | C 1s | O 1s | N 1s | S 2p | P 2p | W 4f | Co 2p |
| MWCNT | 98.9 | 1.1 | - | - | - | - | - |
| MWCNT_N8 | 97.8 | 1.1 | 1.1 | - | - | - | - |
| MWCNT_N8_Co4 | 95.6 | 3.5 | 0.5 | - | 0.1 | 0.2 | 0.1 |
| GF | 95.9 | 4.1 | - | - | - | - | - |
| GF_N8 | 96.5 | 2.4 | 1.1 | - | - | - | - |
| GF_ND8 | 97.3 | 1.6 | 1.1 | - | - | - | - |
| GF_NS8 | 97.6 | 1.4 | 0.7 | 0.3 | - | - | - |
| GF_N8_Co4 | 96.7 | 2.3 | 0.7 | - | - | 0.2 | 0.1 |
| GF_ND8_Co4 | 95.7 | 2.8 | 1.0 | - | 0.1 | 0.3 | 0.1 |
| GF_NS8_Co4 | 95.6 | 3.1 | 0.5 | 0.3 | 0.1 | 0.3 | 0.1 |

[a] Determined by the areas of the respective bands in the high-resolution XPS spectra.

As can be seen in Table 1, the presence of the heteroatoms N and S in the doped materials is confirmed, indicating that the doping procedures were successful, and the composite materials atomic percentages also reveal that the POM atoms are present, albeit in relatively small amounts. There is also an oxygen percentage increase after the POM immobilization, asserting the POM presence in the composite materials. A noteworthy change after POM immobilization is the decrease in the N1s atomic percentage in the composite materials. Considering that XPS is a surface technique that analyzes depths up to 10 nm, the presence of the POM at the surface of the materials may have hindered the nitrogen detection. The N1s high-resolution XPS spectra of the N-containing prepared materials are shown in Figure 4, and the obtained relative atomic percentages of nitrogen in different chemical environments are presented in Table 2. The XPS N1s spectra of the N-containing prepared materials were deconvoluted into three main peaks, which were assigned to pyridinic N (398.5 eV), pyrrolic N (400.1 eV), and quaternary N (401.6 eV) [45–47]. In the case of materials MWCNT_N8 and GF_ND8, a fourth peak at 404.1 eV was found and attributed to nitrogen oxide and/or nitrate species [45]. It is possible to observe that after the POM immobilization, the deconvolution of the peak assigned to quaternary N is not possible for all composite materials. This may be explained by the decrease in signal-to-noise ratio in the POM-containing materials, and by the fact that the analyzed photoelectrons originate from deeper layers within these materials, as the downward slope to lower binding energies of the background signal shows, which is not present before the POM immobilization. This also shows that the POM is present at the surface of the N-containing carbon materials, hindering the N1s photoelectron detection, as stated previously.

**Table 2.** Relative atomic percentages of nitrogen presented in the XPS high-resolution N1s spectra of the prepared carbon materials.

| Material | % N | | | |
|---|---|---|---|---|
| | 398.5 eV (Pyridinic N) | 400.1 eV (Pyrrolic N) | 401.6 eV (Quaternary N) | 404.1 eV (N-Oxides) |
| MWCNT_N8 | 44.0 | 25.5 | 17.3 | 13.2 |
| MWCNT_N8_Co4 | 50.1 | 49.9 | - | - |
| GF_N8 | 56.9 | 31.7 | 11.4 | - |
| GF_ND8 | 45.3 | 29.8 | 13.3 | 11.6 |
| GF_NS8 | 67.9 | 32.1 | - | - |
| GF_N8_Co4 | 63.4 | 36.6 | - | - |
| GF_ND8_Co4 | 65.3 | 34.7 | - | - |
| GF_NS8_Co4 | 66.1 | 33.9 | - | - |

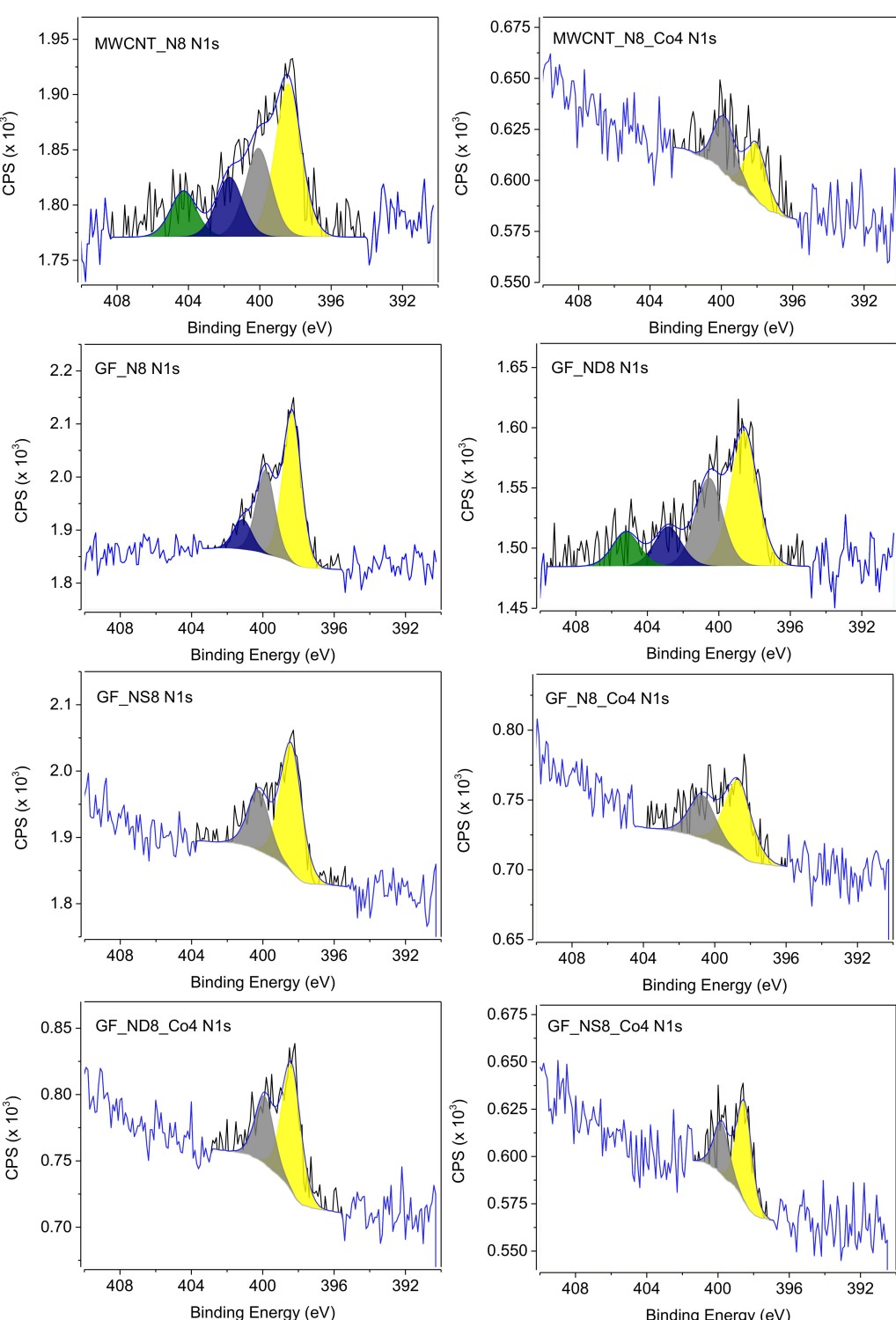

**Figure 4.** Deconvoluted N1s high-resolution spectra of MWCNT- and GF-based materials (yellow: pyridinic N; gray: pyrrolic N; blue: quaternary N; green: N-oxides/nitrates).

The C 1s high-resolution spectra of all materials are shown in Figure S3, and they were deconvoluted as follows: a main peak at 284.6 eV assigned to sp$^2$ C, which is characteristic of graphitic structures; a peak at 285.2 eV corresponding to the sp$^3$ C hybridization; a peak at 286.9 eV ascribed to C in C-O-C; a peak at 288.2 eV assigned to C in C=O; a peak at 289.3 eV corresponding to C in O-C=O; and a peak at 291.0 eV, attributed to $\pi-\pi$* transitions [48]. In all heteroatom-doped materials, there are contributions of C-N and C-S moieties, but these

have binding energies in the 285.2–286.9 eV range, which overlap with the peaks at 285.3 and 286.9 eV, rendering their deconvolution impossible [49,50]. However, in Table S2, it is possible to observe an increase in the relative atomic percentages of these peaks for all the heteroatom-doped materials when compared to the pristine carbon materials, indicating the presence of such contributions. In the O1s high-resolution spectra of all materials (shown in Figure S4), it is possible to identify a peak at 530.7 eV, which is associated with O in C=O and COOH groups, and a second peak at 532.8 eV, which is attributed to O in C-OH groups [51,52]. After the doping procedures, it is possible to observe a decrease in the peaks at 532.8 eV, due to the annealing step, and after the POM immobilization, there is a clear increase in the same peak for all composite materials. As in the C1s spectra, there is an undistinguishable overlap between the contributions of O in C-OH groups from the carbon materials and O present in the POM in the composite materials O1s spectra. The S2p high-resolution spectra (shown in Figure S5) of the sulfur-containing materials revealed two peaks at 164.1 and 165.3 eV, which were associated with the S2p$^{3/2}$ and S2p$^{1/2}$ doublet in thiophene-S ($\Delta E$ = 1.2 eV) [53,54]. The P2p high-resolution spectra (shown in Figure S6) present doublets at binding energies of 133.6 eV and 134.6 eV, corresponding to the 2p$^{3/2}$ and 2p$^{1/2}$ contributions, respectively. In the case of material GF_N8_Co4, it was not possible to deconvolute the P2p spectra due to the low signal-to-noise ratio. In the W 4f high-resolution spectra (Figure S7), the peaks can be resolved as 4f$^{7/2}$ and 4f$^{5/2}$ doublets that appear at 35.5 eV and 37.7 eV, respectively. For cobalt, only the 2p$_{3/2}$ region was analyzed (Figure S8), and the peak can be observed at 781.9 eV [35].

### 3.1.4. Scanning Electron Microscopy

The morphology of two of the samples was evaluated by SEM, and Figure 5 shows the SEM images for GF_ND8 and GF_ND8_Co4 at 50,000× *g* magnification. In Figure 5a, the folded graphene sheets are clearly observed, while in Figure 5b, these are covered with aggregates that correspond to the POM clusters.

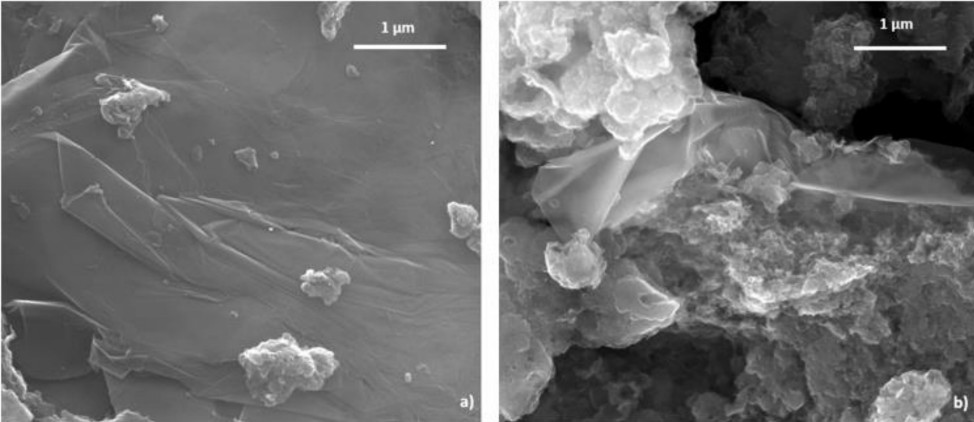

**Figure 5.** SEM images of GF_ND8 (**a**) and GF_ND8_Co4 (**b**) at 50,000× *g* magnification.

To ensure that the observed aggregates corresponded to POM clusters, the backscattered electron (BSED) mode with elemental mapping was used (Figure 6), as it gives elemental information, since it is based on the atomic number Z [55]. In BSED, the brighter sites represent a higher concentration of elements with a high atomic number, in this case tungsten, and it corresponds well with the mapping images. It can also be noticed that the distribution of tungsten (POM) is heterogeneous throughout the sample. The elements Co and P were not able to be mapped for the same time of acquisition as W due to their lower concentration in the sample.

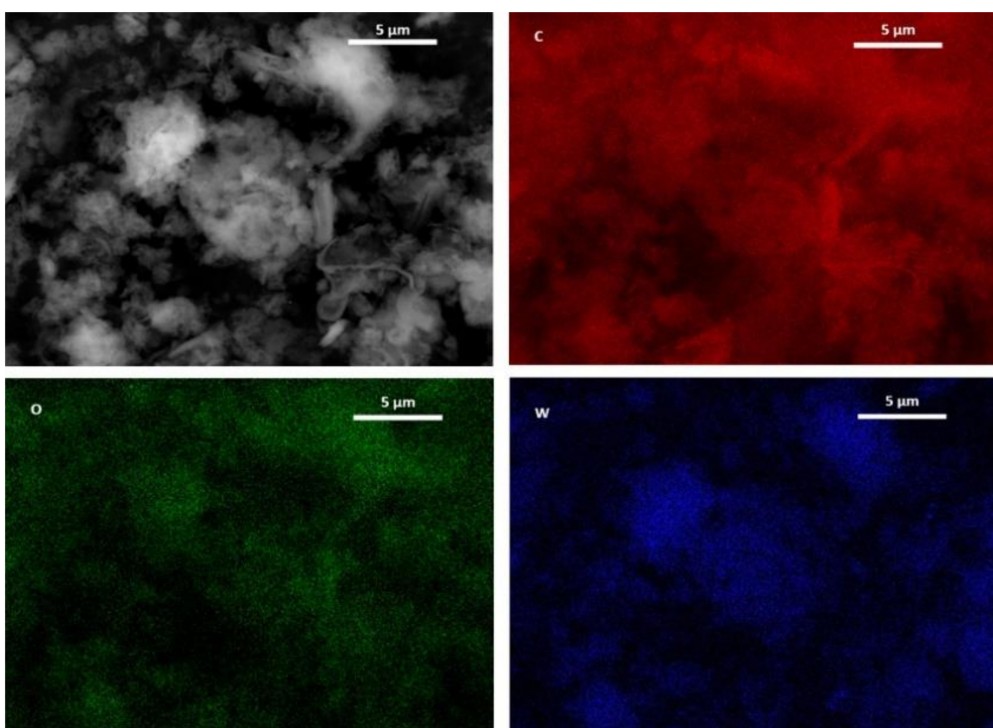

**Figure 6.** SEM and EDX elemental mapping images of GF_ND8_Co4 at 10,000× *g* magnification.

### 3.2. Electrocatalytic Performance

3.2.1. Oxygen Reduction Reaction

Initially, the ORR electrocatalytic performances of MWCNT_N8, GF_N8, GF_ND8, and GF_NS8 were assessed by CV in KOH saturated with nitrogen and oxygen. The CVs of the four doped-carbon nanomaterials are presented in Figure S9, where it can be clearly observed that in nitrogen-saturated electrolyte, no peak can be detected, whereas in oxygen-saturated electrolyte, all doped-carbon materials present an irreversible reduction peak, corresponding to oxygen reduction, at $E_{pc}$ = 0.79, 0.78, 0.77, and 0.70 V vs. RHE for MWCNT_N8, GF_N8, GF_ND8, and GF_NS8, respectively. Pt/C, pristine MWCNT, and GF were also evaluated in the same experimental conditions presenting the ORR peak at $E_{pc}$ = 0.55, 0.64, and 0.86 V, respectively (data not shown).

Further evaluation of the ORR electrocatalytic performances of the prepared doped-CM were conducted by LSV in KOH saturated in both $N_2$ and $O_2$. The LSVs at 1600 rpm of all doped-CM as well as pristine MWCNT, GF, and Pt/C are shown in Figure 7a. It is important to note that these LSVs correspond to those in $O_2$-saturated KOH after subtraction of the blanks (corresponding LSVs in $N_2$-saturated KOH). All the main ORR parameters are presented in Table 3. The $E_{onset}$ can be determined by different methods [4,56,57], and here, we considered the one that assumes it as the potential at which the ORR current is 5% of the diffusion-limiting current density. The results show that all doped-carbon materials presented similar $E_{onset}$ values ranging from 0.75 to 0.85 V vs. RHE, with MWCNT_N8 presenting the closest value to the one obtained for Pt/C (0.92 V vs. RHE). Moreover, MWCNT_N8 presented the highest $j_L$ value ($-4.0$ mA cm$^{-2}$) compared with the other doped ECs where $-2.9 \geq j_L \geq -2.4$ mA cm$^{-2}$. In the same condition, Pt/C presented a $j_L = -4.3$ mA cm$^{-2}$. It can be clearly seen that for both the MWCNT and GF, the doping process led to an improvement of ORR electrocatalytic activity owing to the presence of nitrogen atoms, which is in agreement with several works that have reported the enhancement of ORR electrocatalytic activity after the N-doping of carbon materials [58,59]. After doping, the $E_{onset}$ value of MWCNT shifted 16 mV to more positive potentials, and the $j_L$ value doubles. For graphene, the impact is not so significant with an increase between 21 and 45%.

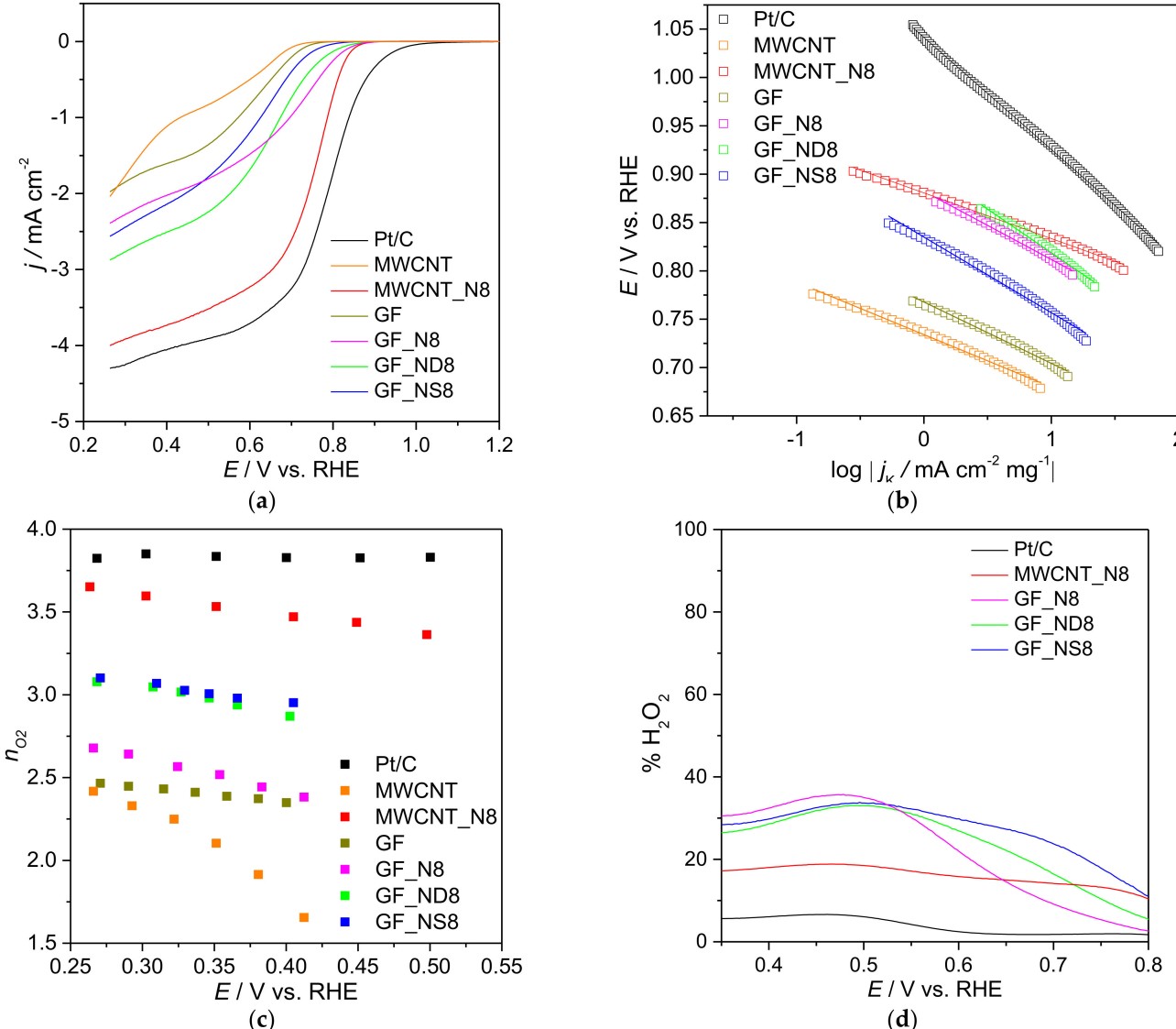

**Figure 7.** ORR LSV curves obtained in KOH (0.1 mol dm$^{-3}$) saturated with O$_2$ for Pt/C, MWCNT, MWCNT_N8, GF, GF_N8, GF_ND8, and GF_NS8 at 1600 rpm and 0.005 V s$^{-1}$ (**a**), the corresponding ORR Tafel plots (**b**), $n_{O_2}$ at several potential values (**c**), and the estimated percentage of H$_2$O$_2$ produced (**d**).

The introduction of heteroatoms in the carbon matrix generally creates different charges or introduces spin distribution of the sp$^2$ carbon plane, which enables the adsorption and activation of oxygen molecules, promoting its reduction. From all the heteroatoms generally used to dop carbon materials, nitrogen has proven to be the most promising one to improve their ORR performances. Considering the XPS results in Table 1, we cannot directly relate the better performance of MWCNT_N8 with the N%, as all the doped-carbon materials present 1.1% of N except for GF_NS8 (0.7%). Pyridinic N atoms, due to the electron donated to the conjugated p bond of graphene and to the lone pair of electrons, are known to facilitate reductive oxygen adsorption and thus have been reported to behave as electroactive sites for ORR [60,61]. Quaternary N is also known to be an electroactive site for ORR [60,62]. As for the total percentage of nitrogen, the better performance of MWCNT_N8 cannot be directly related to the percentages of pyridinic or graphitic N atoms (Table 2). These results suggest that other factors may have an influence on the ORR performance.

The kinetics of ORR were evaluated through the estimation of the slopes (Table 3) of Tafel plots, which can be observed in Figure 7b. All doped materials present lower Tafel slopes than Pt/C, suggesting that oxygen molecules can be easily adsorbed and activated at their surfaces. In addition, the obtained Tafel slope values between 47 and 90 mV dec$^{-1}$ suggest that the conversion of MOO- (intermediate surface-adsorbed species) to MOOH (M is an empty site on the electrocatalyst surface) rules the global reaction rate [63].

**Table 3.** Onset potentials ($E_{\text{onset}}$), diffusion-limiting current density values ($j_{\text{L}}$, at 1600 rpm), and Tafel slopes and determined from the ORR LSV curves in $O_2$-saturated 0.1 M KOH and the mean value of number of electrons transferred for $O_2$ molecule ($n_{O_2}$).

| Sample | $E_{\text{onset}}$ (5% Total) | $j_{\text{L}}$ (mA cm$^{-2}$) | Tafel (mV dec$^{-1}$) | $n_{O_2}$ |
|---|---|---|---|---|
| Pt/C | 0.92 | −4.30 | 116 | 3.8 |
| MWCNT | 0.69 | −2.04 | 54 | 2.1 |
| MWCNT_N8 | 0.83 | −4.00 | 47 | 3.5 |
| GF | 0.72 | −1.98 | 63 | 2.4 |
| GF_N8 | 0.83 | −2.39 | 70 | 2.5 |
| GF_ND8 | 0.80 | −2.88 | 90 | 3.0 |
| GF_NS8 | 0.75 | −2.56 | 78 | 3.0 |
| MWCNT_N8_Co4 | 0.85 | −3.52 | 41 | 3.5 |
| GF_N8_Co4 | 0.83 | −2.88 | 50 | 3.5 |
| GF_ND8_Co4 | 0.85 | −3.18 | 90 | 3.7 |
| GF_NS8_Co4 | 0.84 | −3.03 | 40 | 3.2 |

The number of electrons transferred per $O_2$ molecule ($n_{O_2}$) were estimated applying the Koutecky–Levich equation to the LSVs acquired at different rotation speeds ranging from 400 to 3000 rpm. Figure 7c shows the $n_{O_2}$ values vs. the applied potential and Figures S10 and S11 show the K-L plots of Pt/C and pristine materials and for doped-CM, respectively. The K-L plots of all materials except pristine MWCNT present parallel lines with similar slopes between 0.25 and 0.55 V vs. RHE, suggesting that the number of electrons transferred per $O_2$ molecule does not change much with the applied potential. For Pt/C, a mean value of 3.8 was obtained, while for pristine MWCNT and GF, the values were 2.1 and 2.4, respectively. In alkaline medium, the ORR process proceeds though a direct (four-electron) or an indirect (two-electron) pathway. In the first, $O_2$ is directly reduced to $H_2O/HO^-$, whereas in the second, oxygen is initially reduced to $HO_2^-$, and then, the intermediates are reduced to $H_2O/HO^-$ [4]. These results suggest that in the potential range scanned, Pt/C was involved in a direct process, while pristine MWCNT and GF were involved in an indirect process with the formation of hydrogen peroxide. The doping procedure led to significant changes with MWCNT_N8 reaching an $n_{O_2}$ of 3.5 and GF_ND8 and GF_NS8 a $n_{O_2}$ = 3.0, suggesting that MWCNT_N8 is involved in a direct process while the other two are involved in a mixed regime. For GF_N8, no significant changes were observed when compared with pristine GF.

To confirm the results obtained for the doped-CM and Pt/C, RRDE measurements were performed. The estimated percentages of $H_2O_2$ produced were calculated as detailed in the supporting information file, and the results are presented in Figure 7d. Pt/C presented a low $H_2O_2$ percentage (7%), which corresponds well with the $n_{O_2}$ = 3.8 obtained. The MWCNT_N8 also showed relatively low % $H_2O_2$ (19%) when compared with the doped graphene materials with $H_2O_2$ percentages of 35.9%, 33.3%, and 33.6% for GF_N8, GF_ND8, and GF_NS8, respectively. These values seem to be in accordance with the *n* values estimated from the K-L plots. Still, care should be taken with direct comparison between these two methods, as both present limitations [64–67]. First, the oxidation of $H_2O_2$ on Pt is not a mass-transfer limited process, and second, the electrocatalysts prepared present a rough and heterogeneous structure that may change the geometry of the electrode and

introduce turbulence in the electrolyte flow, leading to estimated $n_{O_2}$ values that may not reflect the real ORR electrocatalytic performance.

To further improve the doped-CM, these were then modified with sandwich-type phosphotungstate $[Co_4(H_2O)_2(PW_9O_{34})_2]^{10-}$ (Co4), and their ORR performance was evaluated in the same experimental conditions. Even though linker molecules are usually introduced to the carbon nanostructures to promote the POM attachment to CM, this strategy was not followed in this work, as this may lead to an increase in electrical resistance of the composite and an unwanted decrease on the electrocatalytic activity [68].

The CVs of the new composite materials are depicted in Figure S12 and, as for the doped-CM, the CVs in $N_2$ do not present any peak, while in $O_2$-saturated KOH, all composite materials present an irreversible ORR peak at $E_{pc}$ = 0.80, 0.75, 0.80, and 0.78 V vs. RHE for MWCNT_N8_Co4, GF_N8_Co4, GF_ND8_Co4, and GF_NS8_Co4, respectively. The LSVs in $O_2$-satrurated KOH are shown in Figure 8a, and we can clearly see some changes when compared with doped-CM. All composites present very similar $E_{onset}$ values ranging from 0.83 to 0.85 V vs. RHE, but now, their $j_L$ values are closer. There was a slight decrease for MWCNT_N8_Co4 (from $-4.0$ to $-3.5$ mA cm$^{-2}$), while for the other three composites, there was an improvement (see Table 3). This behavior is likely because the active sites arising from the POM immobilization do not compensate in the case of MWCNT_N8_Co4 for the loss of N-induced active sites.

The number of electrons involved was also estimated from KL plots (Figure S13) and $n_{O_2}$ = 3.5, 3.5, 3.7, and 3.2 were obtained for MWCNT_N8_Co4, GF_N8_Co4, GF_ND8_Co4, and GF_N8_Co4, respectively. As above, the immobilization of Co4 on MWCNT_N8 does not produce any advantage, while for the other composites, there is an improvement in particular for GF_ND8_Co4. As before, to confirm these results, RRDE measurements were performed. The estimated percentages of $H_2O_2$ produced were $\approx$22%, 23%, 18%, and 32% for MWCNT_N8_Co4, GF_N8_Co4, GF_ND8_Co4, and GF_NS8_Co4, respectively. These values follow the same trend as the number of electrons involved with GF_ND8_Co4 presenting the highest decrease in the percentage of $H_2O_2$ produced from 33% ($n_{O_2}$ = 3.0) to 18% ($n_{O_2}$ = 3.7).

The results obtained in terms of $E_{onset}$, $j_L$, and number of electrons transferred are comparable with several reported results for other Co-POM or Co-containing compounds immobilized on carbon materials, as it can be observed in Table S3.

All the results discussed above were based on the LSV plots involving current densities per nominal area, $j_L$, which referred to the geometric area of the electrode; however, the direct comparison of the nominal current density does not completely describe the effect of the POM deposition on the intrinsic ORR activity of these materials. To discard the influence of surface areas both from the support and composite, the current densities were normalized to the corresponding double-layer capacitances (Figure S14), which were considered as an approximated estimation of the electrochemically active surface areas (ECSAs).

The proportional relationship between ECSA and the double-layer capacitance ($C_{dl}$) along with the similar nature of the materials evaluated in this work makes it possible to compare the materials' $C_{dl}$ values. The double-layer capacitance values were determined via charging tests consisting of CV measurements at increasing scan rates (see full details in the Supplementary Materials file and Figures S15–S17). So, $C_{dl}$ values were calculated from the slopes of the linear fittings of CV current densities (measured at the same potential of 0.95 V vs. RHE, $j_{0.95}$) reached at different scan rates (Figure S18) and are depicted in Table 4. Still, these calculated capacitance values must be seen as estimated values due to the existence of some faradaic contributions in the CV plots of the charge–discharge tests. In this context, the $C_{dl}$ can be considered as an estimation of the number of accessible electrocatalytically active sites for a particular electrocatalyst [69]. For all Co4@doped-CM composites, there is a decrease in $C_{dl}$ value in comparison with the doped carbon materials. This behavior suggests that the dopant moieties are probably located on the carbon materials surfaces, making them susceptible to be easily coated by the POM clusters, hindering their exposure to the electrolyte, and therefore reducing their electroactive surface areas.

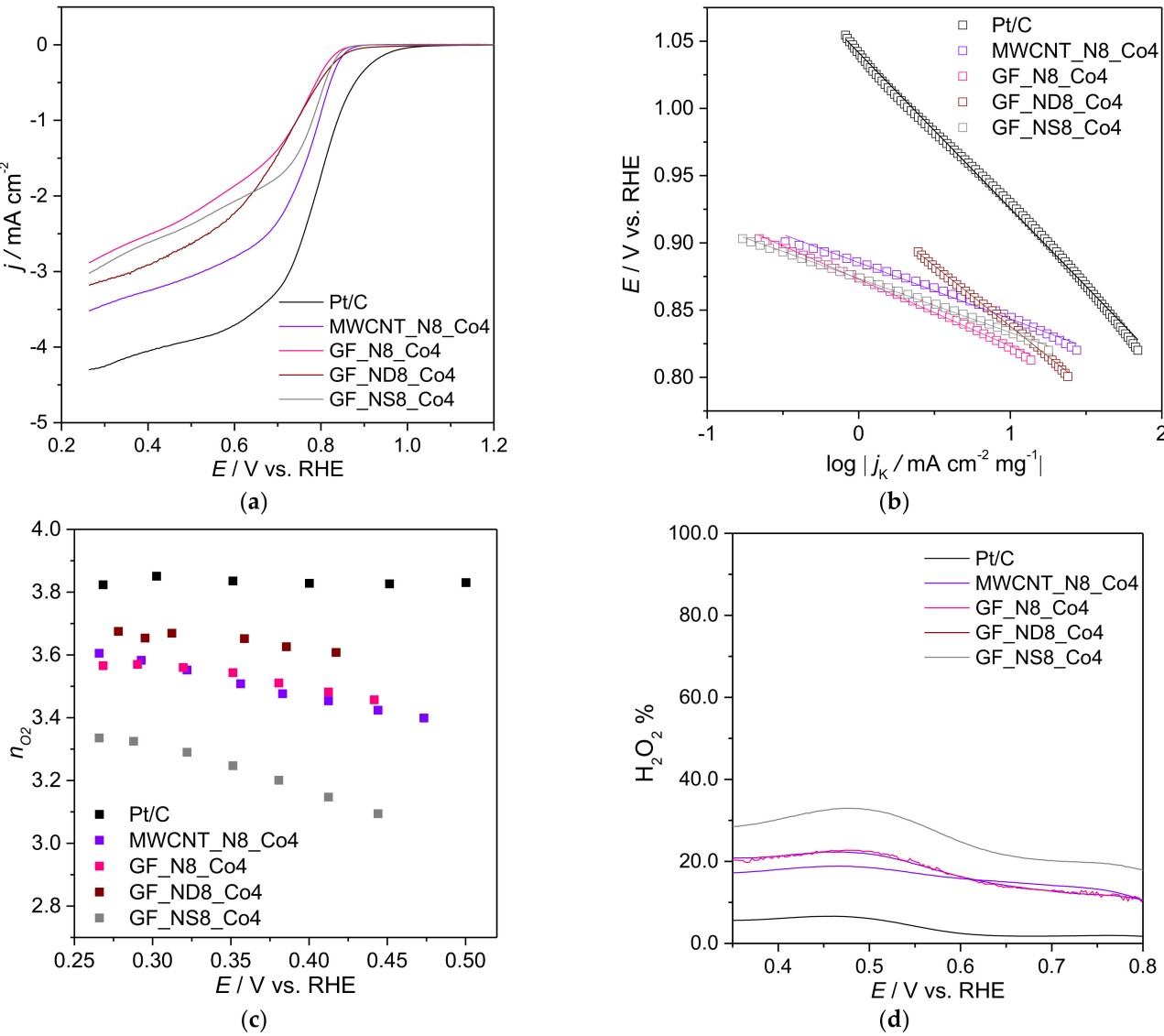

**Figure 8.** ORR LSV curves obtained in KOH (0.1 M) saturated with $O_2$ for Pt/C, MWCNT_N8_Co4, GF_N8_Co4, GF_ND8_Co4, and GF_NS8_Co4 at 1600 rpm and 0.005 V s$^{-1}$ (**a**), the corresponding ORR Tafel plots (**b**), $n_{O_2}$ at several potential values (**c**), and the estimated percentage of $H_2O_2$ produced (**d**).

**Table 4.** Electrocatalytically active surface area (ECSA) values of the doped-CM and the corresponding Co4@doped-CM.

| CM Support | $C_{dl}$ [a]/ mF cm$^{-2}$ | Composite | $C_{dl}$ [a]/ mF cm$^{-2}$ | Composite/CM Support $C_{dl}$ Ratio |
|---|---|---|---|---|
| MWCNT_N8 | 0.0122 | MWCNT_N8_Co4 | 0.0116 | 0.95 |
| GF_N8 | 0.0141 | GF_N8_Co4 | 0.0059 | 0.42 |
| GF_ND8 | 0.0081 | GF_ND8_Co4 | 0.0051 | 0.63 |
| GF_NS8 | 0.0190 | GF_NS8_Co4 | 0.0050 | 0.26 |

[a] Double-layer capacitance values used as ECSA estimations.

As it can be observed in Figure S14, the electroactive surface area has a huge impact on the ORR performance. The increase in the ORR activity of GF-based materials after POM immobilization is much more impressive especially for GF_N8_Co4 ($j_L$ increase of 188%)

and GF_NS8_Co4 ($j_L$ increase of 346%). These studies also revealed that even though the results are good, they are still far from that obtained for Pt/C.

Another relevant parameter that was subject to investigation was the tolerance of the electrocatalysts to methanol crossover. In methanol-based fuel cells, fuel crossover from the anode to the cathode may occur and hence reduce cathodic performance, if electrocatalysts are sensitive to it. As such, tolerance to methanol was evaluated using chronoamperometric tests lasting 2500 s, at 1600 rpm and at $E$ = 0.46 V vs. RHE. At the 500 s mark, 2 mL of methanol was injected in the electrolyte (0.1 mol dm$^{-3}$ KOH). These results are collected in Figure 9a. As it can be observed, Pt/C has a current drop of 48%, while both sets of materials (doped-CM and respective composites) present current retention percentages between 79 and 90%. Even though Pt-based materials have better ORR performance than most electrocatalysts, they have the disadvantage of being highly reactive to the methanol oxidation reaction. This affects its ORR activity performance, lowering the obtained current density.

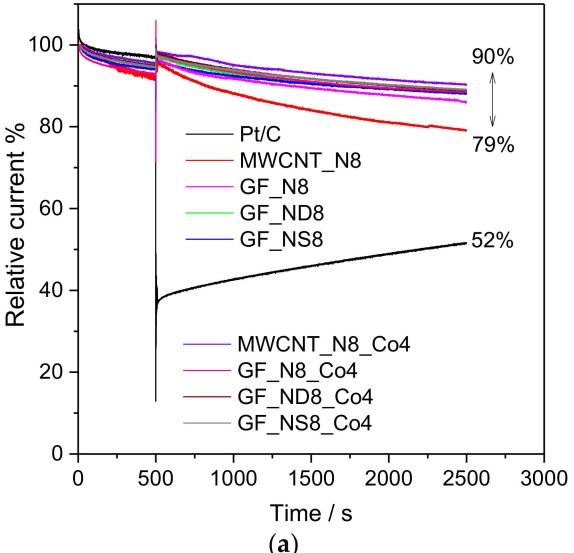
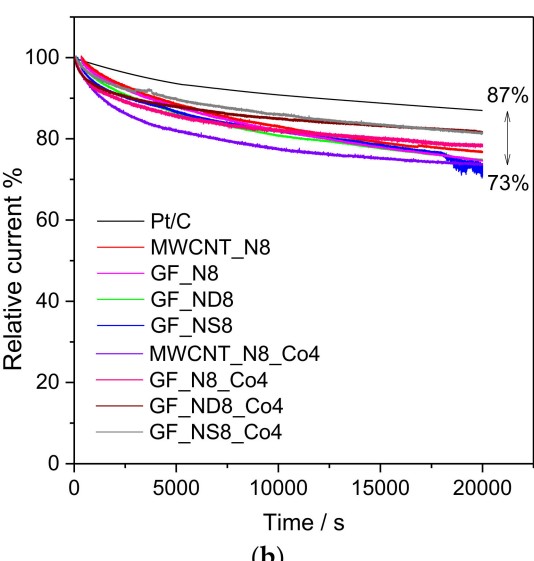

**Figure 9.** Chronoamperometric responses of the prepared electrocatalysts with the addition of 0.5 mol dm$^{-3}$ methanol after ≈ 500 s, at $E$ = 0.46 V vs. RHE, at 1600 rpm, in 0.1 mol dm$^{-3}$ O$_2$-saturated KOH (**a**); Chronoamperometric responses at $E$ = 0.46 V vs. RHE, at 1600 rpm, in 0.1 mol dm$^{-3}$ O$_2$-saturated KOH for 20,000 s (**b**).

Stability is also a crucial parameter while assessing the electrocatalytic activity of ORR electrocatalysts. The electrocatalysts stability was assessed by CA at $E$ = 0.46 V vs. RHE for 20,000 s in oxygen-saturated alkaline electrolyte, and the results of both sets of materials are presented in Figure 9b. The Pt/C electrocatalyst shows a good stability by retaining 87% of its initial current density after 20,000 s, while the other electrocatalysts showed slightly lower current retentions with values ranging from 73% (GF_NS8) to 82% (GF_ND8_Co4 and GF_NS8_Co4).

Even though previous studies by our group with this POM [35] have shown that carbon materials have a crucial role protecting the Co4 from decomposition at high pH values, inductively coupled plasma optical emission spectrometry (ICP-OES) analysis of the electrolyte was performed to evaluate the possible leakage or decomposition of the electrocatalyst. The results showed that no cobalt or tungsten leakage into the electrolyte is observed after chronoamperometric tests, reinforcing our previous findings.

### 3.2.2. Oxygen Evolution Reaction

The OER electrocatalytic performance of the composite materials was also evaluated in alkaline media, and the LSVs can be observed in Figure 10a. One of the parameters that

is commonly determined to evaluate the OER electrocatalysts performance is the potential that is needed to reach $j = 10$ mAcm$^{-2}$, which is a value corresponding to the current density anticipated at the electrode in a solar water-splitting device (under sunlight) with an efficiency of 10% [4]. Thus, generally, the overpotential ($\eta_{10}$) at $j = 10$ mAcm$^{-2}$ is taken as a reference point, and the $\eta_{10}$ values obtained for the Co4 composites are depicted in Table 5, along with the current densities produced ($j$).

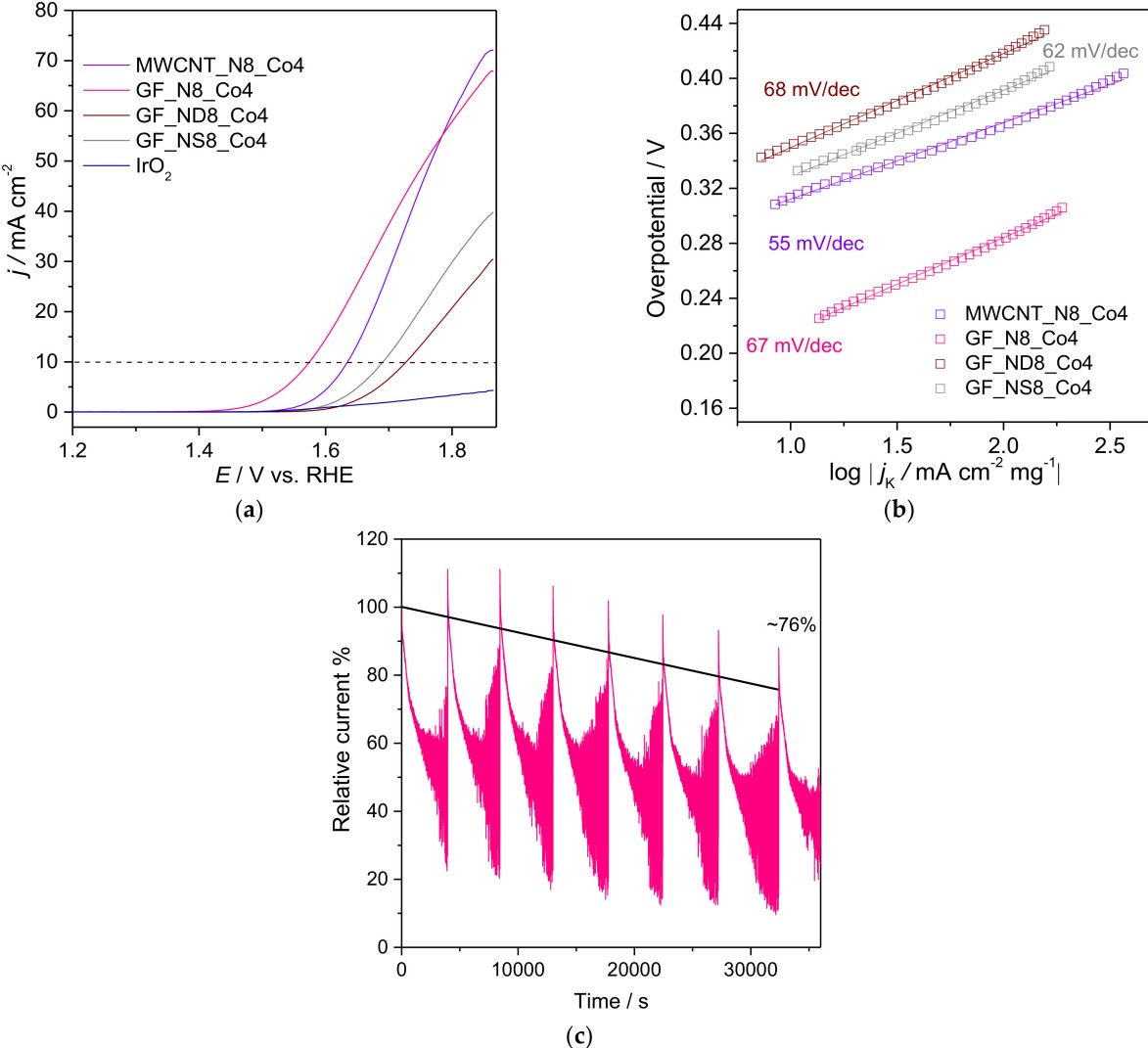

**Figure 10.** OER LSV curves obtained in KOH (0.1 M) saturated with N$_2$ for MWCNT_N8_Co4, GF_N8_Co4, GF_ND8_Co4, and GF_NS8_Co4 at 1600 rpm and 0.005 V s$^{-1}$ (**a**), the corresponding OER Tafel plots (**b**), and the chronoamperometric responses in 0.1 mol dm$^{-3}$ O$_2$–saturated KOH for 36,000 s and at 1600 rpm (**c**).

**Table 5.** Overpotential for $j = 10$ mA cm$^{-2}$ ($\eta_{10}$), maximum current density values ($j_{max}$, at 1600 rpm) and Tafel slopes, and determined from the OER LSV curves in N$_2$-saturated 0.1 M KOH.

| Sample | $\eta_{10}$ (V) | $j_{max}$ (mA cm$^{-2}$) | Tafel (mV dec$^{-1}$) |
|---|---|---|---|
| IrO$_2$ | - | 4.31 | 79 |
| MWCNT_N8_Co4 | 0.40 | 72.1 | 55 |
| GF_N8_Co4 | 0.34 | 67.8 | 67 |
| GF_ND8_Co4 | 0.49 | 30.4 | 68 |
| GF_NS8_Co4 | 0.46 | 39.8 | 62 |

The results show that all composites present good OER electrocatalytic activity with GF_N8_Co4 and MWCNT_N8_Co4 presenting the most promising results with $\eta_{10}$ = 0.34 and $\eta_{10}$ = 0.40 V vs. RHE, respectively. The overpotential values for the other two composites (GF_NS8_Co4 and GF_ND8_Co4) were $\eta_{10}$ = 0.46 and $\eta_{10}$ = 0.49 V, respectively. The MWCNT_N8_Co4 and GF_N8_Co4 electrocatalysts also showed the highest current densities at $E_p$ = 1.86 V vs. RHE ($\eta$ = 0.63 V) (72.1 and 67.8 mA cm$^{-2}$, respectively), which were followed by GF_NS8_Co4 with 39.8 mA cm$^{-2}$ and GF_ND8_Co4 with 30.4 mA cm$^{-2}$.

As for ORR, the Tafel slopes were determined using the LSVs from Figure 10a and Tafel plots are shown in Figure 10b. All composites show similar and low Tafel slopes (55–68 mV dec$^{-1}$), suggesting fast kinetics. All these metrics are increasingly close to those collected in the bibliography (obtained under similar testing conditions) for the expensive state-of-art references: RuO$_2$ with $\eta_{10}$ = 0.30 V, Tafel slope = 65 mV dec$^{-1}$, and IrO$_2$ with $\eta_{10}$ = 0.36 V, Tafel slope = 82 mV dec$^{-1}$ [70]. Additionally, our OER results with the IrO$_2$ modified electrode were very far from those reported [11,70,71]. According to the literature, the OER performances of both IrO$_2$ and RuO$_2$ are greatly influenced by the sample preparation method [11,71]. Additionally, as for ORR, the OER performance is greatly affected by the electrocatalysts electroactive surface. So, the OER current densities were normalized to the corresponding double-layer capacitances (Figure S19). After this correction, GF_N8_Co4 confirms its position as the best-performing OER electrocatalyst followed by GF_NS8_Co4. In addition, GF_N8_Co4 performs even better than the state-of-the-art OER electrocatalyst IrO$_2$ after the application of the same correction (see $C_{dl}$ value in Figure S18).

Finally, the stability of the best-performing OER electrocatalyst in alkaline electrolyte, GF_N8_Co4, was assessed via chronoamperometry, and the results are collected in Figure 10c. The plot shows the characteristic local current density drops originated by oxygen bubble formation on the electrode surface, although previous current density values are partially recovered with bubble release. The GF_N8_Co4 electrocatalyst showed relatively good stability with a current retention of 76% after almost 10 h.

## 4. Conclusions

Four composites based on doped-CM and cobalt phosphotungstate POM were successfully prepared by a simple and scalable strategy without the need of linker molecules. Raman and XPS characterization confirmed the MWCNT and GF doping as well as the incorporation of the POM. Furthermore, Raman analysis showed that POM incorporation had a negligible effect on the graphitic structures. All prepared materials (doped–CM and composites) showed electrocatalytic activity toward ORR with GF_NS8 presenting the best performance within the doped–CM considering the current densities normalized by the estimated ECSA. After POM immobilization, the ORR performances of doped-GF materials were improved, increasing the selectivity toward the four-electron process. Additionally, all materials showed good tolerance to methanol presence and good stability. Regarding the OER studies, the most promising material is GF_N8_Co4 with overpotential 0.34 V vs. RHE and $j_{max}$ close to 70 mA cm$^{-2}$, outperforming, in the same experimental conditions, the state-of-the-art IrO$_2$ electrocatalyst.

**Supplementary Materials:** The following supporting information can be downloaded at: https://www.mdpi.com/article/10.3390/catal12040357/s1, Figure S1. FTIR spectra of MWCNT_N8_Co4 (a), GF_N8_Co4 (b), GF_ND8_Co4 (c), and GF_NS8_Co4 (d); Figure S2. FTIR spectra of MWCNT_N8, GF_N8, GF_ND8 and GF_NS8; Figure S3. Deconvoluted C1s high resolution spectra of MWCNT- and GF-based materials; Figure S4. Deconvoluted O1s high resolution spectra of MWCNT- and GF-based materials; Figure S5. Deconvoluted S2p high resolution spectra of GF_NS8 and GF_NS8_Co4 materials; Figure S6. Deconvoluted P2p high resolution spectra of MWCNT_N8_Co4, GF_ND8_Co4, and GF_NS8_Co4 materials; Figure S7. Deconvoluted W4f high resolution spectra of MWCNT_N8_Co4, GF_ND8_Co4, and GF_NS8_Co4 materials; Figure S8. Deconvoluted Co2p high resolution spectra of MWCNT_N8_Co4, GF_N8_Co4, GF_ND8_Co4, and GF_NS8_Co4 materials; Figure S9. CVs of doped-CM/RDE in KOH (0.1 M) saturated in N$_2$ (dash line) and O$_2$ (red line) at 5 mV s$^{-1}$; Figure S10.

ORR polarization curves of Pt/C (a), pristine MWCNT (c) and pristine GF (e) modified electrodes, acquired at different rotation rates in $O_2$-saturated 0.1 mol dm$^{-3}$ KOH solution at 0.005 V s$^{-1}$ and the corresponding Koutecky-Levich (K-L) plots (b, d and f); Figure S11. ORR polarization curves of MWCNT_N8 (a), GF_N8 (c) GF_ND8 (e) and GF_NS8 (g) modified electrodes, acquired at different rotation rates in $O_2$-saturated 0.1 mol dm$^{-3}$ KOH solution at 0.005 V s$^{-1}$ and the corresponding Koutecky-Levich (K-L) plots (b, d, f and h); Figure S12. CVs of Co4@doped-CM/RDE in KOH (0.1 M) saturated in $N_2$ (dash line) and $O_2$ (red line) at 5 mV s$^{-1}$; Figure S13. ORR polarization curves of MWCNT_N8_Co4 (a), GF_N8_Co4 (c) GF_ND8_Co4 (e) and GF_NS8_Co4 (g) modified electrodes, acquired at different rotation rates in $O_2$-saturated 0.1 mol dm$^{-3}$ KOH solution at 0.005 V s$^{-1}$ and the corresponding Koutecky-Levich (K-L) plots (b, d, f and h); Figure S14. ORR LSV curves obtained in KOH (0.1 M) saturated with $O_2$ at 1600 rpm and 0.005 V s$^{-1}$ with current densities normalized to the respective double-layer capacitance values; Figure S15. CVs at different scan rates of doped-CM/RDE in $N_2$-saturated KOH (0.1 M); Figure S16. CVs at different scan rates of Co4@doped-CM/RDE in $N_2$-saturated KOH (0.1 M); Figure S17. CVs at different scan rates of Pt/C and $IrO_2$ in N2-saturated KOH (0.1 M); Figure S18. Current density-scan rate linear fitting plots for all materials. Numeric values correspond to double-layer capacitances ($C_{dl}$) for each material; Figure S19. OER LSV curves obtained in KOH (0.1 M) saturated with $N_2$ at 1600 rpm and 0.005 V s$^{-1}$ with current densities normalized to the respective double-layer capacitance values; Table S1. Samples code throughout manuscript and respective composition; Table S2. Relative atomic percentages of carbon-containing groups from the deconvolution of the C 1s high resolution XPS spectra of the carbon materials; Table S3. Onset potentials ($E_{onset}$), diffusion-limiting current density ($j_L$) and the number of electrons transferred per $O_2$ molecule for carbon-based materials containing Co-POMs or other cobalt materials reported in literature. References [72–74] are cited in the supplementary materials.

**Author Contributions:** The synthesis and characterization of composite materials was performed by N.L., D.M.F. and B.J. helped in the analysis of data. The Raman analysis were conducted by A.J.S.F. The electrocatalytic experiments were conducted by N.L. and I.S.M. D.M.F. and C.F. supervised the research and N.L., B.J., and D.M.F. were responsible for writing of the manuscript. All authors have read and agreed to the published version of the manuscript.

**Funding:** This research was funded by Fundação para a Ciência e a Tecnologia through project PTDC/QUI-ELT/28299/2017 (FOAM4NER) and the APC was funded by the public Budget—OE by Fundação para a Ciência e a Tecnologia, through the project UIDB/50006/2020 and UIDP/50006/2020.

**Acknowledgments:** DF acknowledge the FCT/MCTES by the work contracts (in the scope of the framework contract foreseen in the numbers 4, 5, and 6 of article 23, of the Decree-Law 57/2016, of August 29, changed by Law 57/2017, of July 19) supported by national funds (OE). NL also thanks SERP+ programme and Erasmus Mundus for her grant.

**Conflicts of Interest:** The authors declare that they have no conflict of interest.

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
