# Peer review of "Cobalt Phosphotungstate-Based Composites as Bifunctional Electrocatalysts for Oxygen Reactions"

_catalysts, doi:10.3390/catal12040357_

Round 1
Reviewer 1 Report
The present manuscript reports the preparation, characterization and application of four composites based on doped carbon materials and cobalt-phosphotungstate as ORR and OER electrocatalysts in alkaline medium (pH=13). The authors investigated the ORR performance of the composites which showed onset overpotentials in the range of 0.83 to 0.85 V vs. RHE, stability over 20000 seconds and selectivity of O2 reduction. The composites were characterized by FT-IR, Raman, XPS and SEM. The manuscript is in general well written and referenced.
I have a few points for the authors to consider.
- Abstract: The authors mention that the composites are stable over 20000 seconds. It would be useful for the readers if the authors mention the number of catalytic cycles as well.
- The authors mentioned in the introduction, background information of POM chemistry. There are a lot more recent review articles that might want to consider: eg. Chem. Soc. Rev., 2012, 41, 7403–7430; Chem. Soc. Rev., 2014, 43, 5679.
- Page3, line 122: Please report the amounts used in mmols as well.
- Page 5, line 170: Please define the ID and IG parameters and their ratio
- The authors report the IR spectra of the POM modified composite and the POM on its own. I would suggest the authors adding also the IR of the MWCNT_N8 composite before the addition of the Co-POM.
The present manuscript presents a family of composite materials with interesting catalytic function for the ORR and OER. I am happy to recommend publication of the manuscript after minor revision.
Reviewer 2 Report
In the present work, the authors have explored carbon materials (CM) and cobalt-phosphotungstate (MWCNT_N8_Co4, GF_N8_Co4, 19 GF_ND8_Co4, and GF_NS8_Co4) materials as ORR, OER catalyst. The produced catalyst materials have been characterized in detail by using various techniques such as FTIR, Raman, XRD, SEM, and also electrochemically. The authors report significant improvement in electrocatalytic performance for composite electrodes. The work is concise and well-written; however, some points (listed below) are not clear to the reviewer.
1.Throughout the manuscript, we encounter typos, misuse of an upper case within the sentences. Some of the mistakes are listed below
-upper case in the experimental (...., Sodium tungstate dehydrate......)
-the electrode suffered a cleaning process (in line 142)9
- withing (line 498) and isGF_N8_Co4 (line 502)
2.The authors have prepared and investigated different samples; however, it is hard to follow the coding of the samples. It will be easier to understand for the readers if the authors present the samples and their content as a separate table in the experimental or result/discussion section.
3.XPS peak fittings in P2p and Co2p regions (Figure S5 and S7) are not convincing, the peak intensities are poor—mainly noise-like signal. Have the authors considered the error % in their fitting? Can the authors give Co % and P % measured by XPS?
4.In the current version, the comparison of the electrodes` performance with the literature is missing. The authors should add the missing information.
Reviewer 3 Report
The article entitled «Cobalt Phosphotungstate based composites as bifunctional electrocatalysts for oxygen reactions» is devoted to the topical issue of obtaining effective catalysts that do not contain precious metals for fuel cells and electrolyzes. The article presents extensive studies of the composition and structure of a number of doped carbon materials and composites based on them. The activity of the obtained materials in ORR and OER was studied in detail. The article presents original research, well-written and detailed, has sufficient novelty. All this allows us to recommend the article for publication in the Catalysts magazine after a slight revision.
It is necessary to justify the choice of an alkaline medium for studying the catalysts activity. The fact is that proton-exchange membranes are commercially available, while effective anion-exchange membranes do not exist today.
For carbon carriers, including commercial carriers, BET surface area data must be added.
It is necessary to add XRD data for carbon supports before and after doping, this can add useful data on the materials structure. XRD must be added for composite materials MWCNT_N8_Co4, GF_N8_Co4, GF_ND8_Co4, GF_NS8_Co4.
What is the error in determining the concentration of elements according to XPS. For example, the N concentration for the MWCNT_N8 sample is 1.1%, and the N spectrum is highly noisy. In this case, not only the concentration of N is determined, but also the proportion of various forms of nitrogen (Table 2).
Table 3 does not indicate at what potential the number of electrons (n) was determined.
In table 5, you need to add data on IrO2.
Round 2
Reviewer 2 Report
The authors have carefully addressed all concerns raised by the reviewer. I recommend this manuscript be accepted.